# Status of leptoquark models after LHC Run-2 and discovery prospects at future colliders

Nishita Desai[*]

*Department of Theoretical Physics,*

*Tata Institute of Fundamental Research,*

*Mumbai, India 400005*

Amartya Sengupta

*Department of High Energy Physics and Cosmology,*

*State University of New York, Buffalo, USA[†]*

We study limits from dilepton searches on leptoquark completions to the Standard Model in the parameter space motivated by anomalies in the $b \to s$ sector. After a full Run-2 analysis by LHCb, the disparity in lepton flavour universality violation has disappeared. However, the mismatch in angular distributions as well as in $B_s \to \mu^+\mu^-$ partial width is still unresolved and still implies a possible new physics contribution. We probe three models of leptoquarks — scalar models $S_3$ and $R_2$ as well as vector leptoquark model $U_1$ using non-resonant dilepton searches to place limit on both the mass and couplings to SM fermions. Current limits on leptoquarks with both non-uniform or uniform coupling to lepton flavours are calculated. Interestingly, if leptoquark couplings to electrons and muons are indeed universal, then the $U_1$ model parameter space that corresponds to the anomalous contribution should already be accessible with Run-2 data in the non-resonant $e\mu$ channel. In the non-universal case, there is a significant exclusion in couplings, but not enough to reach regions that explain observed anomalies. We, therefore, examine the prospective sensitivity at the HL-LHC as well as of a 3 TeV future muon collider. For the vector leptoquark model, we find that a muon collider can probe all of the relevant parameter space at 95% confidence with just 1 fb$^{-1}$ data whereas $R_2$ and $S_3$ models can be excluded at 95% with 5 fb$^{-1}$ and 6.5 fb$^{-1}$ luminosity respectively.

---
[*] nishita.desai@tifr.res.in

[†] amartyas@buffalo.edu

# I. INTRODUCTION

An exciting development in recent years has been the measurement of ratios of decay widths in the semileptonic rare decays of B-mesons [1–4], hinting at lepton flavour-universality violation (LFV). The latest of these [1, 2] showed a measurement consistent with the SM for certain lepton universality, however, there remains a mismatch with the measured branching fraction of $B_s \to \mu^+\mu^-$ [3, 4] and in the angular distribution in the decay $B \to K^*\mu^+\mu^-$ [5, 6]. Unsurprisingly, this has led to a spirited effort to understand the source of the mismatch with the predictions of the Standard Model (SM) and to provide new physics explanations for it. In particular, there have been several dedicated studies that determine global fits to data in terms of effective field theoretic operators (see e.g. [7–13]). There has also been some effort to explain the anomalies in terms of new particles, notably with new vector bosons or leptoquarks [14–38]. The effects of the presence of such new particles can generally be seen in other observables besides the LFV ratios, and in particular, in the high energy tails of certain distributions observable at the LHC. In this paper, we examine the expected effects of leptoquarks with minimally required properties to cause observed anomalies in the B-sector and report on current constraints and future prospects of their detection.

We start by providing a bare-bones introduction to how the Effective Field Theoretic (EFT) framework is used and translated to the measurement of the high-energy observables that we examine in this paper. EFT provides a useful method to describe the low-energy physics processes in which the short-distance (i.e. high-energy or UV) physics is encapsulated in the Wilson coefficients whilst the rest of the long-distance physics is expressed in terms of effective operators with those having dimensions higher than four being suppressed by powers of an energy scale to maintain the mass dimension of each term in the Lagrangian. The analytic form of the Wilson coefficient can then be calculated by "matching" the expressions calculated from the EFT with the expressions from the full UV theory. We can use the published value by one of the multiple groups to translate the B-meson observations into best-fit values of the appropriate Wilson coefficients [7–11, 39]. We then match these values to the expressions derived from the leptoquark model under study and study the consequence of what that means on other production mechanisms at the LHC.

The anomalies seen in the data fall into two categories — (1) in the neutral current sector

with $b \to s$ transitions, and (2) in the charged current sector with $b \to c$ transitions. In this work, we concentrate mainly on models that explain the first of these [40], however, it is known that one of the models we study viz. the $U_1$ vector leptoquark can explain both simultaneously(see e.g. table 2. of [14])

The relevant observations that motivate this work based on the full Run 1 and 2 dataset are shown in table II in the appendix. For completeness, we show both the pre-December 2022 LHCb announcement [1, 2] numbers, as well as the latest measurements.

The low-energy effective theory for the $b \to s$ flavour changing neutral current sector is described in terms of an effective Hamiltonian which can be written as

$$\mathcal{H}_{\text{eff}} = -\frac{4G_F}{\sqrt{2}} V_{tb} V_{ts}^* \left\{ \sum \mathcal{C}_i(\mu) \mathcal{O}_i(\mu) \right\}$$

where $\mathcal{C}_i(\mu)$ are the Wilson coefficients. The effective operators relevant to our study are

$$\mathcal{O}_9^{l_1 l_2} = \frac{e^2}{(4\pi)^2} (\bar{s}\gamma_\mu P_L b)(\bar{l}_1 \gamma^\mu l_2), \quad \mathcal{O}_{10}^{l_1 l_2} = \frac{e^2}{(4\pi)^2} (\bar{s}\gamma_\mu P_L b)(\bar{l}_1 \gamma^\mu \gamma^5 l_2) \tag{I.1}$$

Multiple fitting studies have found that the operator whose Wilson coefficient shows significant deviation from the predicted SM value is the $C_9$ and that the most likely discrepancy seems to be in the $C_9^{\mu^+ \mu^-}$ coefficient. To stay consistent with the latest data, we use the

| Author (Year) | Model Dependent | Data Driven |
|---|---|---|
| Ciuchini et al (2022) [7] | $[-1.25, -0.72]$ | $[-1.10, 1.05]$ |
| Ciuchini et al (2019) [8] | $[-1.37, -1.05]$ | $[-1.47, -0.93]$ |
| Algueró et al (2019) [9] | $[-1.15, -0.81]$ | |
| Alok et al (2019) [10] | $[-1.27, -0.91]$ | |
| Hurth et al (2021) [11] | $[-1.07, -0.83]$ | |

TABLE I. Best Fit values for the new physics contribution to the operator $C_9$. The first of these contains the updated 2022 results. The fits taking into account angular distributions still favour a similar range as before the 2022 LHCb data release even though the overall best fit $1\sigma$ range is now consistent with the SM value of zero.

most recent best-fit results as reported in [7]. We shall use the best-fit values that correctly give the angular correlations as well (the so-called "model-dependent" fit). However, later in the paper when we examine future prospects, we also show the overlap with the fully agnostic data-driven fits. For an overview of the best-fit $C_9$ values see table I. Currently, we proceed by using the value

$$C_9^{\mu^+\mu^-} = -0.98 \pm 0.27,$$

Multiple studies have also examined the leptoquark UV completion and calculated explicit expressions for $C_9^{\mu^+\mu^-}$ from each model. In this work, we use these expressions to investigate the LHC constraints on the couplings and mass of the leptoquarks. We make only the minimal assumptions, i.e. only the couplings that are necessary to give a contribution to the $b \to s$ anomalies are assumed to be non-zero. As we shall see, in each leptoquark model, the Wilson coefficients $C_{9(10)}$ depend on three parameters roughly as

$$C_9 \sim \left(\frac{y_{22}\ y_{32}}{M}\right)^2$$

where $y_{22}$ is the $s\mu$ coupling, $y_{32}$ is the $b\mu$ coupling and $M$ is the mass of the leptoquark. We start by constraining $(y_{22}, y_{32}, M)$ in other production modes without any further assumptions on other leptoquark couplings. This results in the most conservative limits. In the case where there is no LFV, one would expect identical couplings of the leptoquark to electrons, i.e. $y_{22} = y_{21}$ and $y_{32} = y_{31}$. This would also lead to signatures with different flavored dileptons which often have much stronger constraints. These constraints are examined in section III. In the flavour universal case, the strongest limits on leptoquark masses will come from $\mu \to e$ processes including $\mu \to e\gamma$ [41] and $\mu \to 3e$[42] measurements. However, it might be possible that the effects of leptoquarks could be cancelled in loop-induced processes by the presence of other new particles. Studying direct leptoquark production at the LHC allows us to directly probe the lepton-universal case because the observed number of events in $\mu\mu$, $ee$ and $\mu e$ channels will be correlated.

Our paper is structured as follows: we start by listing out the model Lagrangian and the resulting Wilson coefficients for $C_9$ in section II. We then examine the current LHC constraints in various search channels in section III and expected detection prospects of future colliders are calculated in section IV.

## II.  LEPTOQUARK MODELS

Leptoquarks are bosons which carry both $SU(2)_L$ and colour $SU(3)$ charges and therefore couple to both leptons and quarks. Given that we need to get the right contribution to $C_9^{\mu^+\mu^-}$, this corresponds to a leptoquark that at a minimum couples to muons and to $b$ and $s$ quarks. There are three known leptoquark models that give the right kind of contribution [14–16, 43], which we describe below. We use the standard names for the fields, viz. $S_3$, $R_2$ and $U_1$ and the numbers in brackets that follow correspond to (n-plet of SU(3), n-plet of SU(2), $U(1)_Y$ hypercharge). Of these, $S_3$ and $R_2$ are scalar fields and $U_1$ is a vector field.

### A.  Scalar Leptoquark $S_3$ $(\bar{3}, 3, 1/3)$

The first leptoquark model we consider is $S_3(\bar{3}, 3, 1/3)$ which is a $SU(2)_L$ triplet of scalar leptoquark states with hypercharge $1/3$. $S_3$ is the only scalar leptoquark model that can simultaneously predict $R_K^{exp} < R_K^{SM}$ and $R_{K^*}^{exp} < R_{K^*}^{SM}$ at tree level [44–47]. The Lagrangian for the $S_3$ model is

$$\mathcal{L}_{S_3} = y_L^{ij}\bar{Q}_i^C i\tau_2(\tau_k S_3^k)L_j + h.c., \tag{II.1}$$

where $Q_i$ and $L_j$ are $SU(2)_L$ doublet fermion fields corresponding to quarks and leptons of the $i^{\text{th}}$ ( $j^{\text{th}}$) generation respectively, $\tau_k$ are the generators of $SU(2)_L$, and $y_L^{ij}$ stands for a Yukawa matrix for the left-handed fermions. The three triplet component states of $S_3$ carry charges $Q = -2/3, 1/3$ and $4/3$ respectively. Expanding out the $SU(2)_L$ components and referring to the leptoquarks as $S_3^Q$, we get

$$\begin{aligned}
\mathcal{L}_{S_3} = &-y_L^{ij}\bar{d}_{Li}^C \nu_{Lj}S_3^{1/3} - \sqrt{2}y_L^{ij}\bar{d}_{Li}^C \ell_{Lj}S_3^{4/3} \\
&+ \sqrt{2}(V^*y_L)^{ij}\bar{u}_{Li}^C \nu_{Lj}S_3^{-2/3} - (V^*y_L)^{ij}\bar{u}_{Li}^C \ell_{Lj}S_3^{1/3} + h.c.,
\end{aligned} \tag{II.2}$$

of which only the $\bar{d}_{Li}^C \ell_{Lj}S_3^{4/3}$ term contributes to $O_9$. One can extract the Wilson coefficients for the $b \to sl^-l^+$ decay [14–16, 43],

$$C_9^{\ell_1\ell_2} = -C_{10}^{\ell_1\ell_2} = \frac{\pi v^2}{V_{tb}V_{ts}^*\alpha_{em}}\frac{y_L^{b\ell_1}(y_L^{s\ell_2})^*}{m_{S_3}^2}, \tag{II.3}$$

## B.  Scalar Leptoquark $R_2\,(3, 2, 7/6)$

The second case we consider is a weak doublet of scalar leptoquarks with hypercharge $Y = 7/6$, i.e. $R_2\,(3, 2, 7/6)$.[48] The most general Lagrangian describing the Yukawa interactions with $R_2$ can be written as,

$$\mathcal{L}_{R_2} = y_R^{ij}\bar{Q}_i l_{R_j} R_2 - y_L^{ij}\bar{u}_{R_i} R_2 i\tau_2 L_j + h.c., \tag{II.4}$$

where $y_L$ and $y_R$ are the Yukawa matrices corresponding to left- and right-handed lepton fields respectively. In terms of the components with $R_2^Q$ denoting each leptoquark state with charge $Q$, the Lagrangian can be written as

$$\mathcal{L}_{R_2} = (Vy_R)^{ij}\bar{u}_{Li}\ell_{Rj}R_2^{5/3} + (y_R)^{ij}\bar{d}_{Li}\ell_{Rj}R_2^{2/3}$$
$$+ (y_L)^{ij}\bar{u}_{Ri}\nu_{Lj}R_2^{2/3} - (y_L)^{ij}\bar{u}_{Ri}\ell_{Lj}R_2^{5/3} + h.c. \tag{II.5}$$

The tree-level contribution to the Wilson coefficients $C_9$ through the term $(y_R)^{ij}\bar{d}_{Li}\ell_{Rj}R_2^{2/3}$ amounts to

$$C_9^{\ell_1\ell_2} = C_{10}^{\ell_1\ell_2} = -\frac{\pi v^2}{2V_{tb}V_{ts}^*\alpha_{em}}\frac{y_R^{s\ell_1}(y_R^{b\ell_2})^*}{m_{R_2}^2}, \tag{II.6}$$

## C.  Vector Leptoquark $U_1\,(3, 1, 2/3)$

Finally, we describe the only vector leptoquark model considered in this paper, mainly because it has been the only model that could simultaneously explain both charged current and neutral current anomalies [14]. We consider the $U_1\,(3, 1, 2/3)$ model which gives a single leptoquark state with charge 2/3. The most general Lagrangian consistent with the SM gauge symmetry allows couplings to both left-handed and right-handed fermions, namely

$$\mathcal{L}_{U_1} = \beta_L^{ij}\bar{Q}_i\gamma_\mu L_j U_1^\mu + \beta_R^{ij}\bar{d}_{R_i}\gamma_\mu \ell_{Rj}U_1^\mu + h.c., \tag{II.7}$$

with couplings $\beta_L^{ij}$ and $\beta_R^{ij}$. The contributions to the left-handed couplings to the effective Lagrangian amount to

$$C_9^{\ell_1\ell_2} = -C_{10}^{\ell_1\ell_2} = -\frac{\pi v^2}{V_{tb}V_{ts}^*\alpha_{em}}\frac{\beta_L^{s\ell_1}(\beta_L^{b\ell_2})^*}{m_{U_1}^2}, \tag{II.8}$$

## III. LHC LIMITS

Our goal is to use published LHC data to simultaneously constrain the mass and Yukawa couplings of the leptoquarks. The Wilson coefficient $C_9$ depends on three parameters roughly as

$$C_9^{\ell,\ell} \sim \left( \frac{y_{2\ell}\ y_{3\ell}}{M} \right)^2$$

where $y_{ij}$ refers to the leptoquark coupling between the $i^{\text{th}}$ generation of quark and $j^{\text{th}}$ generation lepton. This corresponds to Yukawa couplings for $S_3$ and $R_2$ models and the gauge coupling for the $U_1$ model. Therefore, it's possible to find a surface in the 3D parameter space that gives the required value of $C_9$. However, most LHC search constraints are in principle only 2D — one coupling that determines the cross-section of the final state and one mass. We, therefore, have several options in which to view the full constraints.

Let us start with $\ell = 2$ (i.e. $\mu$) which contributes to $C_9^{\mu\mu}$. To be able to independently constrain the two Yukawa couplings $y_{22}$ and $y_{32}$, we study three different cases — first setting only $y_{22}$ non-zero (see figure 1, second setting only $y_{32}$ non-zero (see figure 4) and third, setting both equal (see figure 5). Using the upper limits from the non-resonant dimuon search gives us an upper limit on $y_{22}$ at each mass value. It is possible to also determine the minimal allowed value of $y_{22}$ that is consistent with $C_9$ by requiring $y_{32} \leq 1$.

Since the latest LHCb data seem to indicate that electrons and muons have identical behaviour, we can indeed also do a similar exercise with $y_{21}$ and $y_{31}$ which would contribute to $C_9^{ee}$. Besides these, non-zero values of all four couplings (or even a single electron and a single muon coupling) — $y_{21}$, $y_{31}$, $y_{22}$ and $y_{32}$ can give signatures that have differently flavoured leptons in the final state, but without missing energy and therefore with no SM background.

It should be noted that in the case where a single leptoquark state can couple to both electrons and muons, the strongest constraints on couplings and mass of course come from low energy processes in the $\mu \to e$ sector [41, 42, 49]. However, it can still be an interesting exercise to directly probe the case where both $y_{k1}$ and $y_{k2}$ are non-zero. As we see in figure 2, this case is strongly constrained by the LHC, with the $U_1$ model likely to be ruled out already with full run-2 data of 139 fb$^{-1}$.

Since multiple leptoquark states come from the same multiplet, they have identical mass and switching on a single coupling allows the production of multiple states. For calculating

the LHC limits, we allow the production of all leptoquark states and select only that fraction that decays into the final state selected for by the analysis being reinterpreted. For example, in the $S_3$ case, if we look for pair production of leptoquark followed by the decay of each into a muon and a jet by turning on $y_{22} \neq 0$ alone, we allow both the production of pairs of $S_3^{4/3} \rightarrow \bar{s}\mu^+$ as well as pairs of $S_3^{1/3} \rightarrow \bar{c}\mu^+$. Our limits, therefore, are not identical to the simplified model limits that the experimental analysis publishes by producing only one state at a time, with 100% branching fraction into a certain channel. Similarly, when looking at dilepton distributions, we take into account, with interference, all leptoquark states in the t-channel that are allowed by non-zero couplings.

## A. Computational setup

Since we examine the limits from dilepton searches which have been presented in the form of upper limits on generator-level cross sections with fiducial cuts, our computational setup is much simplified. We generate events using `Madgraph5_amc@NLO` [50] with the required fiducial cuts and do not need to perform further detector simulation. This approach has been proven to work well [51] and reproduces expected limits. For the UV models, we use the scalar leptoquark models for $S_3$ and $R_2$ described in [52] and for the vector leptoquark model for the $U_1$ case, we use the model described in [53–55]. When more complicated functionality is required, we use Pythia8 [56] to shower, hadronize and apply the required kinematic cuts on events.

## B. Limits from resonant and non-resonant dilepton searches

We re-interpreted both the dilepton resonance search with 139 fb$^{-1}$ [57] and the non-resonant dilepton search at 139 fb$^{-1}$ [58] from ATLAS. We find that the non-resonant search results in much stronger limits and we continue with this search for the rest of our study. The exclusive dilepton state can only be seen with a t-channel leptoquark exchange. It is possible to have a dilepton plus two jets from strong production of leptoquarks, however, this process is not affected by the leptoquark-fermion couplings (except in the range $y \gtrsim 1$) and results in a fairly fixed mass limit which we deal with in the next subsection. With the interference of SM Drell-Yan production of leptons with the t-channel leptoquark mediated production,

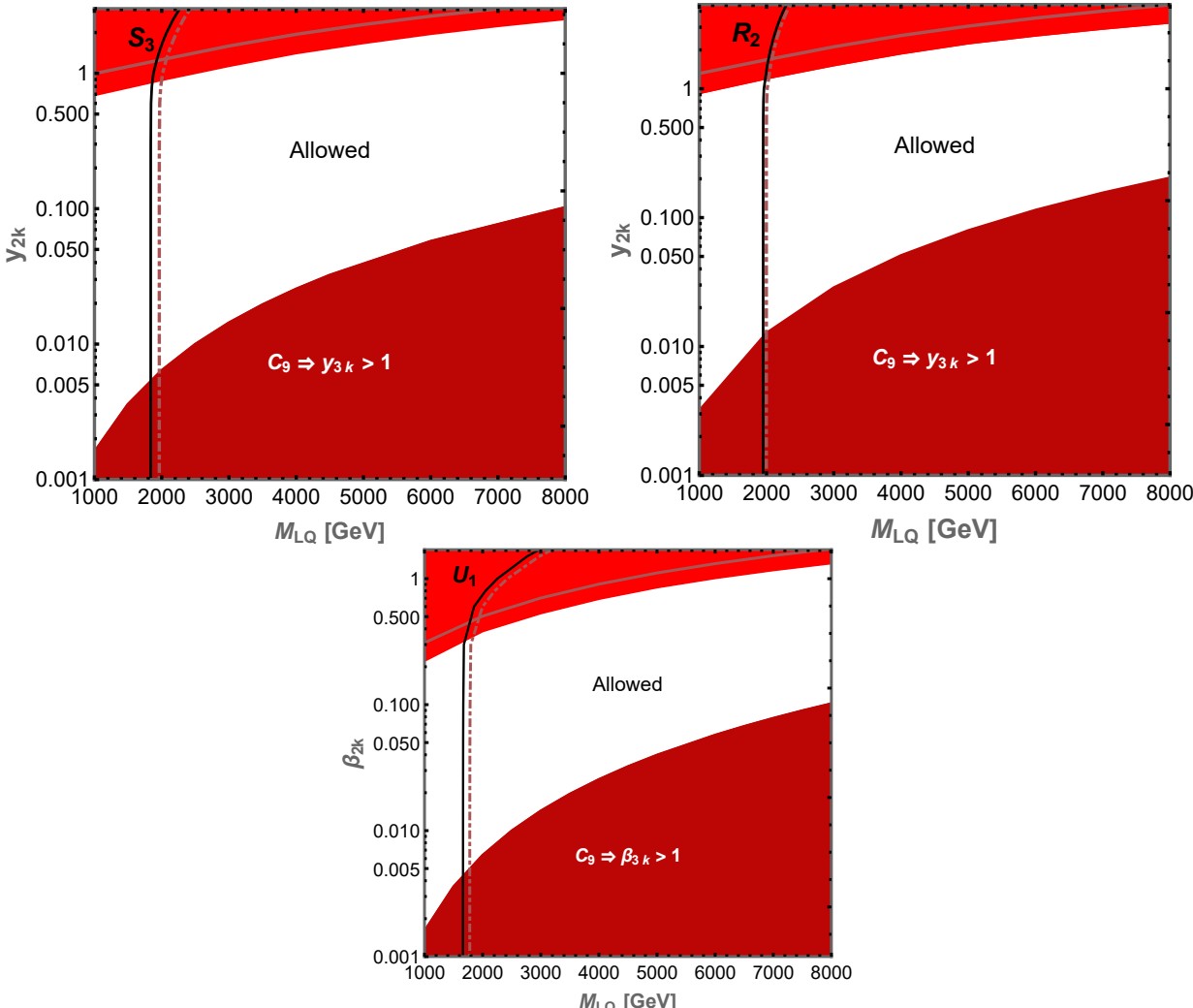

FIG. 1. Exclusion plots $y_{2\ell}$ versus Mass of leptoquark for the $S_3$ (top-left), $R_2$ (top-right) and $U_1$ models (bottom). The bright red regions at the top are disallowed from dimuon searches. The corresponding di-electron limit is the lighter line inside the red region. The solid regions at the bottom are from requiring perturbative couplings consistent with allowed $C_9$. The vertical lines are mass limits from direct leptoquark pair production with the solid line corresponding to second-generation leptons and the dotted corresponding to first-generation. The limits correspond to 139 fb$^{-1}$ data.

one expects to see a change in the shape of the dilepton invariant mass distribution $m_{\ell\ell}$ where $\ell = \mu$ or $e$.

We apply the limits from the ATLAS non-resonant dilepton search by generating events using `Madgraph5_amc@NLO` according to fiducial cuts listed in [58] and using the 95% upper

limits for the most conservative signal region called the "$\mu^+\mu^-$ constructive signal region" (or analogously the $e^+e^-$ constructive signal region). The constructive signal region corresponds to the case where you expect signal events above the EW expectation, which is similar to our case. The experimental analysis uses LO signal shape to model the expected number of events and we therefore also do not use any NLO corrections. The upper limits are provided on the additional cross section above the expected SM Electro-Weak (EW) prediction in the cumulative signal region where $m_{\mu^+\mu^-} \geq 2070$ GeV (or $m_{e^+e^-} \geq 2200$ GeV).

As expected, the effect of having heavy new leptoquarks in the t-channel dies down when either the leptoquark mass is too high or the Yukawa coupling is too small. To account for the interference correctly, we use the difference of the cross-section $pp \to \ell^+\ell^-$ with both leptoquark and EW bosons, and with only EW gauge bosons as our new physics contribution. The result is an excluded region near high Yukawa coupling values, with a larger range, ruled out for smaller leptoquark masses. This is shown as a bright red region in figure 1. The highest allowed value of $y_{2k}$ can be used to further restrict what values of $y_{3k}$ are consistent with $C_9$. A reciprocal analysis done with production via $y_{3k}$ is shown in appendix B. Due to the small bottom fraction in the proton at LHC energies, the Drell-Yan limit is not strong enough to constrain any values of $y_{3k}$ directly.

Currently, there is one different flavour dilepton search [59] performed at 13 TeV, but with only 3.2 fb data analysed. Aside from cuts on $p_T$ of 65 and 50 GeV on electrons and muons respectively, there are requirements that missing energy be less than 25 GeV and $m_T < 50$ GeV to remove contamination from W-boson production which we apply using Pythia 8.3 [56]. The expected background for $m_{e\mu} > 2$ TeV is $0.02 \pm 0.02$. They see one event and interpret it as a statistical fluctuation, setting a limit on a new physics cross-section. We extrapolate the expected limits from this search at 139 fb$^{-1}$. The limits on the $e\mu$ case for the $U_1$ model can be seen in figure 2. The expected background at 139 fb$^{-1}$ is 2.78 events, resulting in an expected 95% upper limit of 0.0185 fb on the production cross section times branching. As can be seen, the $U_1$ model should be completely ruled out with 139 fb$^{-1}$ data. For results in the $e\mu$ channel for $S_3$ and $R_2$ models, refer to appendix C.

### C. Limits from leptoquark-pair production

Direct limits on the mass of the leptoquark based on strong pair-production mode followed by the decay of each leptoquark into a lepton and a jet are presented in [60]. The limits are also presented on generator-level cross-section times branching fraction and can be applied directly to our model. The resulting limit is shown as a solid black vertical line. Since there is no significant improvement in the limit from b-tagging, we use the general lepton+jet limits in all cases. When only $y_{k2}$ is non-zero, i.e. the leptoquark decays to a muon and a jet, we obtain a mass limit for $S_3$ leptoquark at 1840 GeV, for the $R_2$ leptoquark at 1950 GeV and the $U_1$ leptoquark at 1670 GeV. For the case where the leptoquark decays into an electron alone, we get a mass limit for $S_3$ leptoquark at 1964 GeV, for the $R_2$ leptoquark at 1998 GeV and the $U_1$ leptoquark at 1785 GeV. For very high values of the Yukawa ($y > 1$), the limit can be higher by up to 50% depending on the model because of the contribution of the lepton-mediated t-channel diagram. However, the region we are interested in for B-anomalies corresponds to smaller Yukawa couplings which correspond to nearly a constant mass limit. Finally, the larger production cross-section of the vector-like leptoquark doesn't result in comparatively stronger limits because of a reduction in half of the branching fraction once we demand a charged lepton in the decay of both leptoquarks.

There is no published direct limit on the case with an $e\mu$ final state, which if it existed, would give a far better exclusion simply because there is no irreducible SM background and the dominant background would be from mis-identification of leptons.

### D. Missing search: top FCNC decay

Given the need for non-zero leptoquark coupling to the third generation of quarks, this also implies a coupling between the top quark and second-generation leptons for both the $S_3$ and $U_1$ models. In the $R_2$ case, the coupling is either CKM suppressed (in the case of left-handed) or entirely independent and therefore set to zero (in the right-handed case). It would therefore be possible to search directly for FCNC top decay via $t \to c\mu\mu$.

Currently, there are no searches for $t \to c\mu^+\mu^-$ except for a $t \to cZ$ search which requires the dimuon mass to be within 15 GeV of the Z mass [61] and therefore is not directly applicable to our model. A similar measurement from CMS [62] is available from the 8 TeV

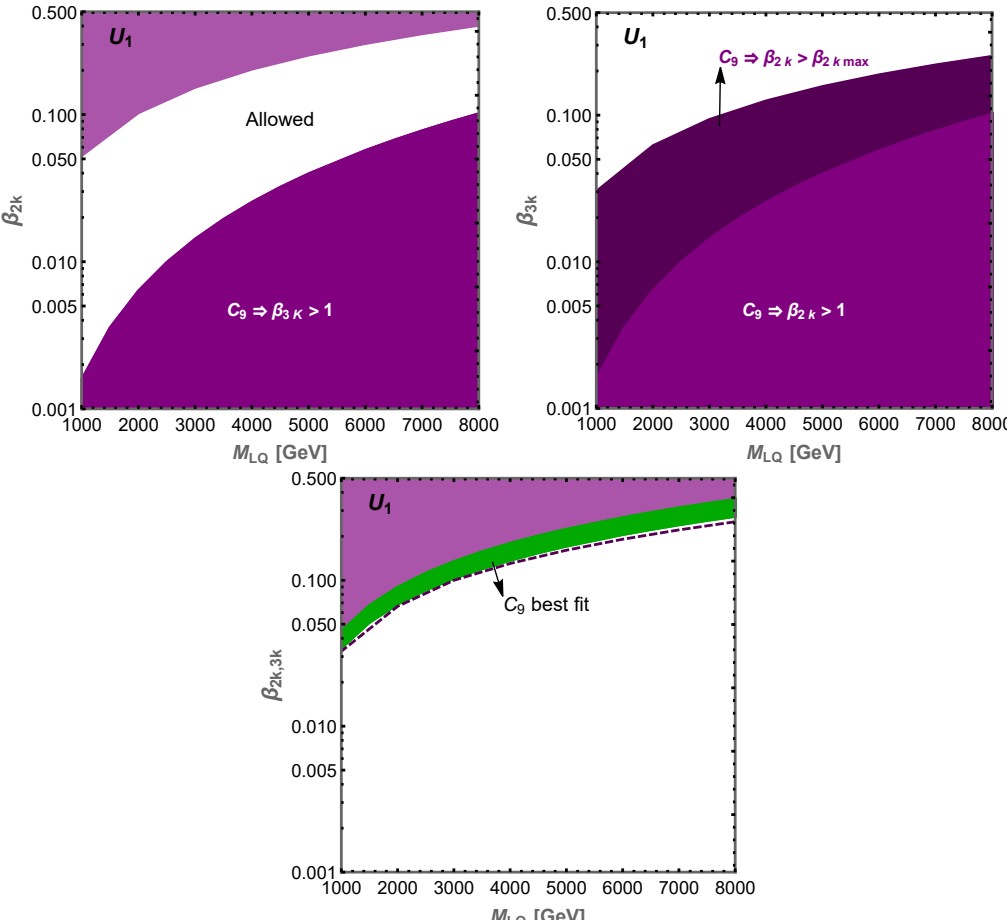

FIG. 2. Limits for the Leptoquark Couplings versus mass for the $U_1$ Model. The dilepton process, in this case, is $pp \to \mu e$ which does not exist in the SM. We, therefore, have strong limits even with 3.2 fb$^{-1}$ data as published in [59]. The top-left panel shows limits on the coupling to second-generation quarks with $y_{22} = y_{21}$, the top-right panel on the coupling to third-generation quarks with $y_{32} = y_{31}$ and the bottom panel shows the case where all four couplings are equal. The green band shows the values corresponding to the best-fit values of $C_9$ The dotted line in this figure shows the expected limit after analysing full 139 fb$^{-1}$ of run-2 data by ATLAS (only partial results is published so far). We see clearly that the universal scenario should be clearly visible with full Run-2 data (and hence is likely already ruled out).

run.

The main background for a $t \to c\mu^+\mu^-$ search is from the SM production of $t\bar{t}\mu^+\mu^-$ via an off-shell Z or $\gamma$ produced in association with $t\bar{t}$. To remove contamination from on-

shell Z, we apply a cut instead $M_{\ell\ell} > 105$ which is outside the Z-mass window selected for by the $t \to cZ$ searches. Assuming the identification acceptances do not change, we can estimate the background for our proposed search using the data driven estimate presented in [61] (denoted by $\sigma_{BG,ATLAS}$). Since we have identical SM production modes for $t\bar{t}Z$ and $t\bar{t}\mu^+\mu^-$, we assume that the generator level transfer factor between these processes is transmitted all the way to the final selection. The kinematic effect of changing the $m_{\ell\ell}$ cut from $|M_{\ell\ell} - M_Z| < 15$ to $M_{\ell\ell} > 105$ can be estimated at generator level and is encapsulated in a single number $f_{\ell\ell}$ Also, we assume that the enhancement in production of $t\bar{t}Z$ in going from 13 TeV to 13.6 TeV ($f_E = \frac{\sigma_{13.6}}{\sigma_{13}}$) remains the same also for $t\bar{t}\mu^+\mu^-$. Thus we have

$$
\begin{aligned}
\sigma_{BG}(\sqrt{s} = 13.6) = \quad & \sigma_{BG,ATLAS} \\
& \times f_E \times f_{\ell\ell} \\
& \times \frac{\sigma(pp \to t\bar{t}\mu^+\mu^-; \sqrt{s} = 13)}{\sigma(pp \to t\bar{t}Z; \sqrt{s} = 13)}
\end{aligned}
\tag{III.1}
$$

Using this, and the expected background cross-section from ATLAS, we calculate an expected background of $7 \pm 2$ events. Given that with the Z-window, the background is estimated at $119 \pm 10$ events, this would correspond to over an order of magnitude improvement in the sensitivity to FCNC branching fraction of the top quark.

## IV. FUTURE PROSPECTS

The best-fit value of the Wilson coefficients for operators that explain the $b \to s$ anomalies suggests a high suppression scale. Using equations (II.6), (II.3) and (II.8), we find that the required scale for both couplings set to one is 16183 GeV for the $R_2$ case and 22887 GeV for the $S_3$ and $U_1$ cases. Naturally, resonantly producing a leptoquark of this mass scale is out of the question at the LHC. We, therefore, investigate both the expected reach of the LHC after the planned high-luminosity run and estimate a conservative reach for a muon collider with CM energy of 3 TeV [63–67]. To illustrate the highest sensitivity case, we choose $y_{22} = y_{32}$ for this calculation. This also allows us to make a comment on the ability of the collider to explore the entire parameter space of interest. A summary of the expected reach of future colliders can be seen in figure 3

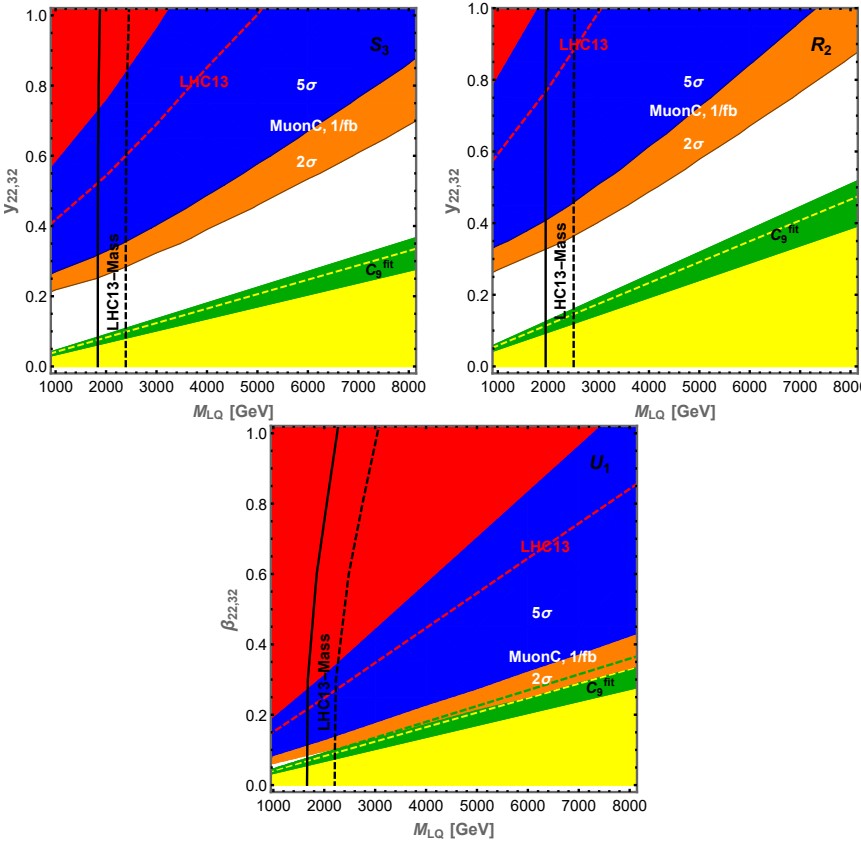

FIG. 3. Current and future reach in leptoquark coupling to muons with leptoquark mass for the $S_3$ Model (top-left), $R_2$ Model (top-right) and $U_1$ Model (bottom). The green region corresponds to the $1\sigma$ region given by global fit $C_9$ values in the model-dependent case whereas the yellow is the data-driven $1\sigma$ region ([7], also see table I). The solid red region is the current 139 fb$^{-1}$ limits with the dotted red line the expected reach after 3 ab$^{-1}$ at the HL-LHC. The solid and dotted vertical lines correspond to mass limits from pair production again corresponding to the 139 fb$^{-1}$ and 3 ab$^{-1}$ luminosity respectively. The blue region corresponds to the parameter space that can be discovered with a $5\sigma$ significance at a 3 TeV muon collider with 1 fb$^{-1}$ whereas the orange region corresponds to the further region that can be probed at 95% confidence at the same collider. The $U_1$ model can be fully excluded with just 1 fb$^{-1}$ data. The $S_3$ and $R_2$ models can also be fully probed with 6.5fb$^{-1}$ and 5fb$^{-1}$ respectively.

## A.  LHC High-Lumi expected limits

Projections for the HL-LHC are made with the luminosity of 3000 fb$^{-1}$. From previous experience, we know that the improvements in limits scale with about the square root of luminosity. Using the expected number of signal and background events for the non-resonant dilepton search, we can probe effects of leptoquarks up to mass 5 TeV for the $S_3$, 3 TeV for the $R_2$ and 9.5 TeV for the $U_1$ model. Conversely, we can probe coupling values as small as 0.4 for $S_3$, 0.55 for $R_2$ and 0.15 for $U_1$ models respectively at 1 TeV leptoquark mass. For comparison, $C_9$ best fit predicts a minimum value of coupling at 0.04, 0.06 and 0.04 for the three models when we set both couplings equal.

The direct search limits from strong production are calculated in a similar way using the published upper limits at 139/fb. We find that the HL-LHC can exclude leptoquark masses of 2.5 TeV for both the $S_3$ and $R_2$ case and 3.0 TeV for the $U_1$ case for the leptoquark decaying into a muon and a jet and 2.6 TeV for both the $S_3$ and $R_2$ case and 3.2 TeV for the $U_1$ case for the leptoquark decaying into an electron and a jet.

## B.  Reach of a Future Muon Collider

Estimating the reach of a future muon collider is more difficult since we do not currently have a detector configuration to be able to simulate a realistic analysis. However, taking lessons from the dilepton and dijet searches at the LHC, we know that a single-bin analysis with a high enough cut on the invariant mass provides a very reliable estimate of reach. We look at $\mu^+\mu^- \to jj$ as our signal. Obviously using b-tagging will be a further improvement that can pinpoint the underlying scenario. However, for this estimate, we just use untagged jets. Given that, acceptance efficiencies of jets are expected to be similar for both signal and background events for a simple dijet search, we proceed with using just generator-level cross sections. A further advantage is the much-reduced probability of extra initial state radiation jets from initial state muons (in sharp contrast to a pp machine).

The main background from the SM comes from the s-channel photon or Z exchange. In the presence of the leptoquark, another Feynman diagram with a t-channel leptoquark exchange needs to be taken into account. We look only at events with $M_{jj} > 500$ GeV. The SM-only cross-section at LO is $5.96 \times 10^{-2}$ pb which corresponds to a statistical error of

about 8 events at a luminosity of 1 fb$^{-1}$. Using this, we can calculate the parameter space corresponding to a $5\sigma$ discovery as well as regions that can be excluded at $2\sigma$. They are shown in figure 3 as blue and orange regions respectively. In the $U_1$ case, we see that a muon collider is capable of excluding the entire viable parameter space with 1 fb$^{-1}$. To exclude the $R_2$ and $S_3$ models would need a luminosity of 6.5 fb$^{-1}$ for $S_3$ and 5 fb$^{-1}$ for R2.

## V.   SUMMARY AND CONCLUSIONS

We examine the limits from direct collider searches on leptoquark models that are capable of explaining the anomalous measurements in the decays of B-mesons. We focus on three specific models — two scalar leptoquark models $S_3$ and $R_2$ and one vector leptoquark model $U_1$. Aside from limits on the mass of the leptoquarks (which can be pair-produced by strong interactions), it is possible to also constrain the couplings to fermions by looking at changes to the shape of the dilepton mass spectrum. Reinterpreting full Run-2 limits from the pair production and non-resonant dilepton searches by ATLAS experiment, we find that current mass limits are 1.84 TeV for $S_3$, 1.95 TeV for $R_2$ and 1.67 TeV for $U_1$ model respectively. We can expect to reach up to 2.5 TeV for $S_3$ and $R_2$ and 3.0 TeV for the $U_1$ respectively with the High-Luminosity LHC run.

Effects of leptoquarks with couplings to muons can potentially be probed in a muon collider. Since there has been considerable interest in a future muon collider recently, we also estimate what the reach of the proposed 3 TeV muon collider would be for the three models in question. We find that with very minimal assumptions, $S_3$, $R_2$ and $U_1$ models show significant deviation in dijet distributions that can be observable for the entire range of interest with less than 6 fb$^{-1}$ data for all three models. It would also be possible to probe the case of universal couplings between $e$ and $\mu$ at the upcoming Electron-Ion Collider (EIC). However, we find that due to low collision energy of this collider, the resultant sensitivity does not compete with LHC limits.

## ACKNOWLEDGEMENTS

ND is supported by the Ramanujan Fellowship grant SB/S2/RJN-070 from the Department of Science and Technology of the Government of India.

————————————————

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

**Appendix A: Relevant observables in the $b \to s$ sector**

| Observable | Experiment | Theory (SM) |
|:---:|:---:|:---:|
| $R_{K_{[0.1,1.1]}}$ | $0.994 \, ^{+0.090}_{-0.082}$ (stat) $^{+0.029}_{-0.027}$ (syst) [2022] [1, 2] | $1.00 \pm 0.01$ [70–72]) |
| $R_{K^*{}_{[0.1,1.1]}}$ | $0.927 \, ^{+0.093}_{-0.087}$ (stat) $^{+0.036}_{-0.035}$ (syst) [2022] [1, 2] | $1.00 \pm 0.01$ [70–72]) |
| $R_{K_{[1.1,6]}}$ | $0.949 \, ^{+0.042}_{-0.041}$ (stat) $^{+0.022}_{-0.022}$ (syst) [2022] [1, 2] | $1.00 \pm 0.01$ [70–72]) |
| $R_{K^*{}_{[1.1,6]}}$ | $1.027 \, ^{+0.072}_{-0.068}$ (stat) $^{+0.027}_{-0.026}$ (syst) [2022] [1, 2] | $1.00 \pm 0.01$ [70–72]) |
| $R_{K^*}^{[0.045,1.1]}$ | $0.66^{+0.11}_{-0.07} \pm 0.03$ [2021] [73] | $0.906 \pm 0.028$ [70–72] |
| $R_{K^*}^{[1.1,6.0]}$ | $0.69^{+0.11}_{-0.07} \pm 0.05$ [2021] [73] | $1.00 \pm 0.01$ [70–72] |
| $R_K^{[1.1,6.0]}$ | $0.846^{+0.042+0.013}_{-0.039-0.012}$ [2021] [74] | $1.00 \pm 0.01$ [70–72] |
| $\mathcal{B}(B_s \to \mu^+\mu^-)$ | $(2.85^{+0.32}_{-0.31}) \times 10^{-9}$ [3, 4] | $(3.66 \pm 0.14) \times 10^{-9}$[75]) |
| $P_5'$ in $B \to K^{(*)} \, l^+ \, l^-$ | [5, 76, 77] | [6, 78] |

TABLE II. A summary of the most relevant experimental results and SM predictions for the observables in $b \to s$ sector.

**Appendix B: Limits on leptoquark couplings to third generation quarks $y_{3k}$.**

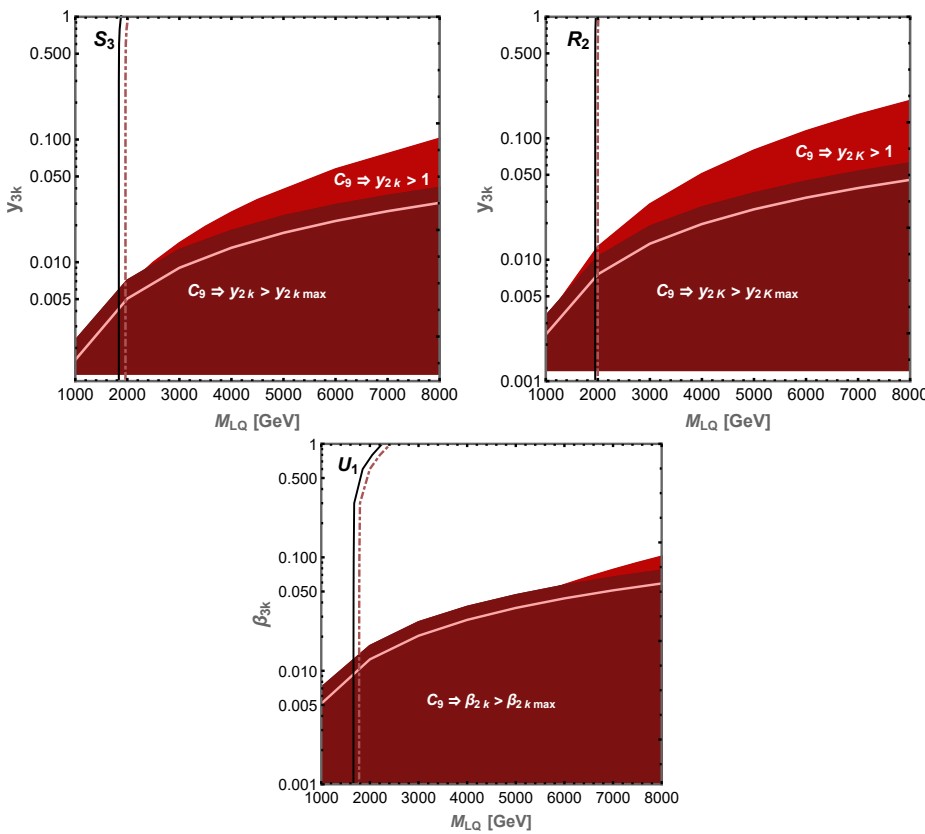

FIG. 4.    Exclusion plots $y_{3\ell}$ versus Mass of leptoquark for the $S_3$ (top-left), $R_2$ (top-right) and $U_1$ models (bottom). The solid regions at the bottom are from requiring perturbative couplings consistent with allowed $C_9$. The darker region is inconsistent with the observed upper limits on $y_{2k}$ in figure 1. The vertical lines are mass limits from direct leptoquark pair production with the solid line corresponding to second-generation leptons and the dotted corresponding to first-generation. The limits correspond to 139 fb$^{-1}$ data.

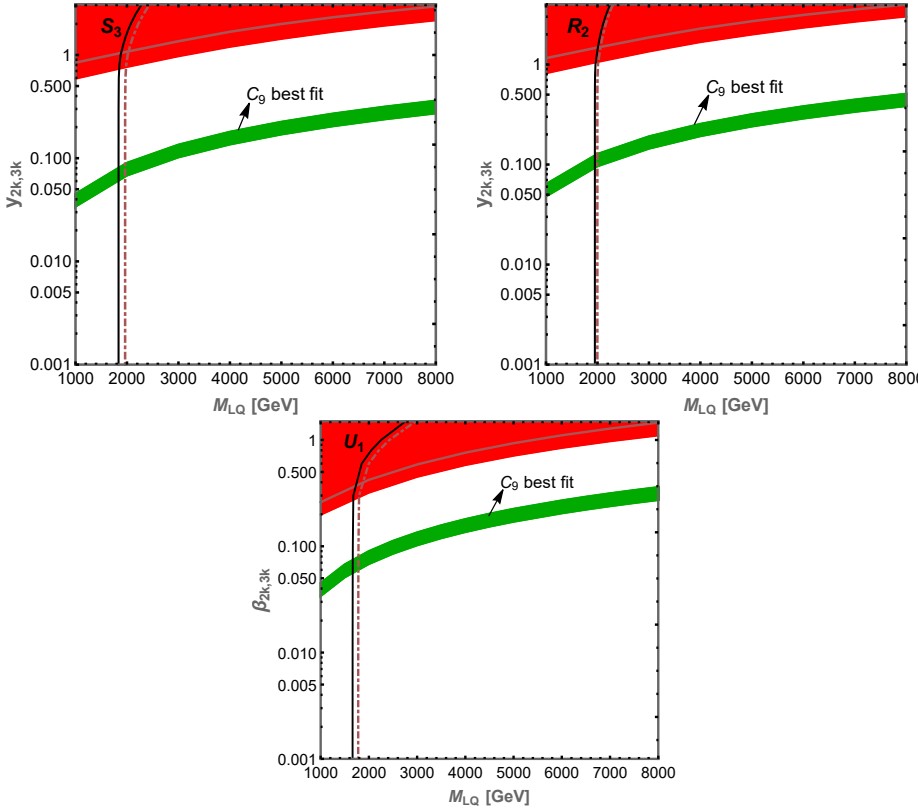

FIG. 5. Exclusion plots in the limited case of $y_{2\ell} = y_{3\ell}$ versus Mass of leptoquark for the $S_3$ (top-left), $R_2$ (top-right) and $U_1$ models (bottom). The solid red region at the top are limits from non-resonant dilepton searches in $\mu^+\mu^-$. The lighter lines inside this region correspond to subleading limits from the similar $e^+e^-$ search. The vertical lines are mass limits from direct leptoquark pair production with the solid line corresponding to second-generation leptons and the dotted corresponding to first-generation. The limits correspond to 139 fb$^{-1}$ data. The green band is the region that corresponds to the coefficient $C_9$ within one sigma of best fit to data.

**Appendix C: Limits on $S_3$ and $R_2$ model parameters in the Lepton Flavour Universal case**

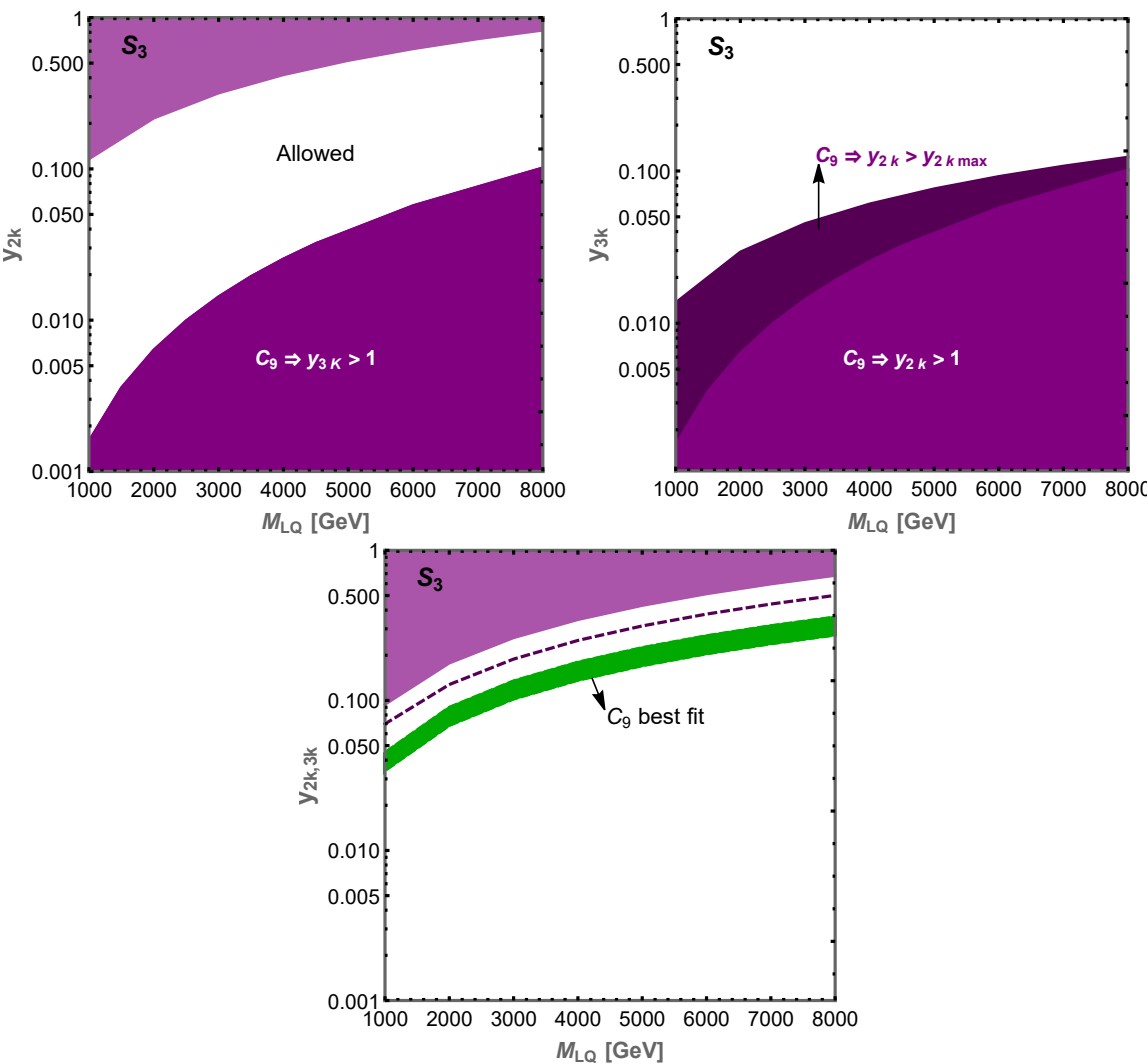

FIG. 6. Limits on the leptoquark couplings via the process $p\,p \to \mu\,e$ in the case of flavour universal couplings to electrons and muons for the $S_3$ Model.

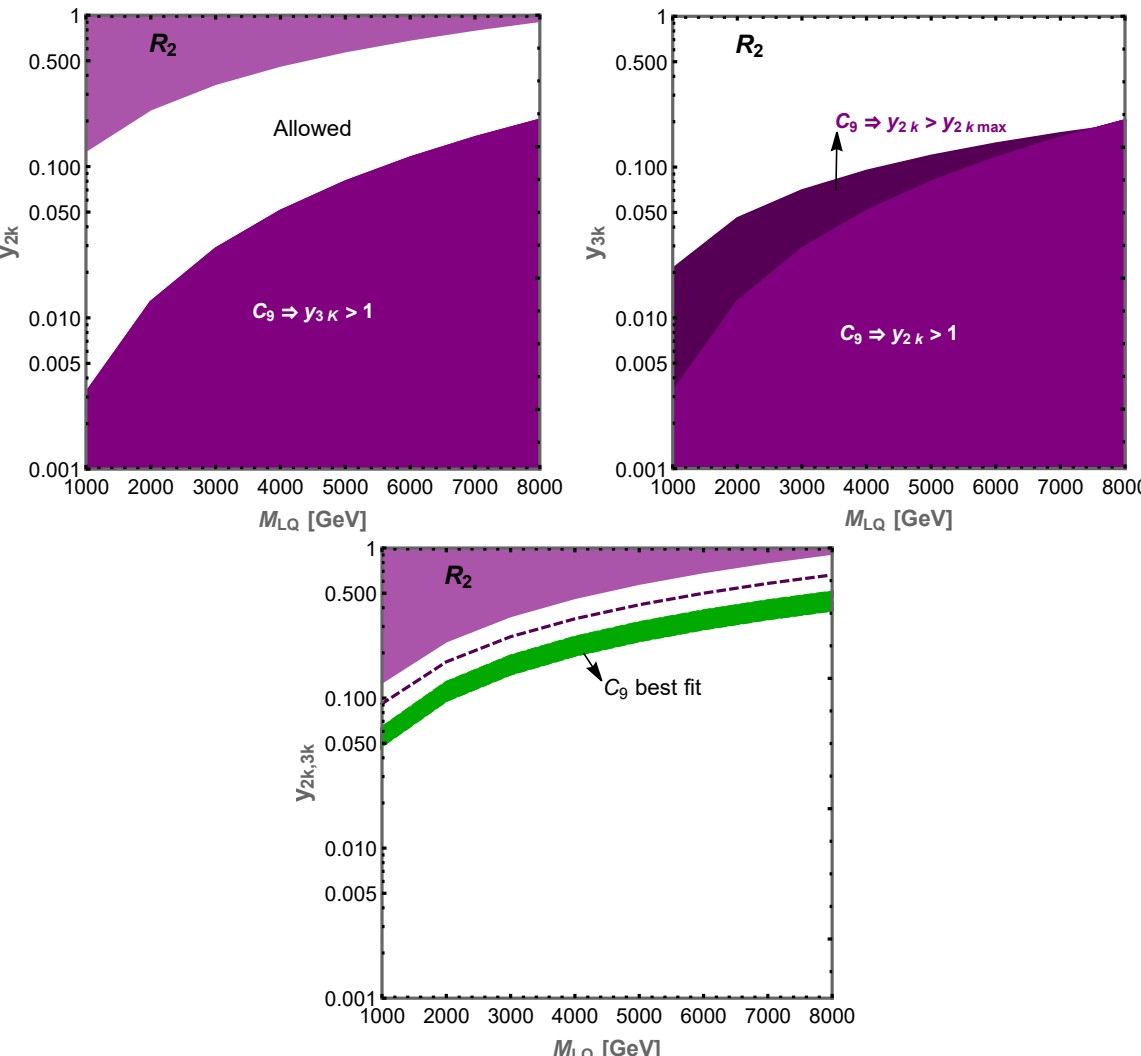

FIG. 7. Limits on the leptoquark couplings via the process $p\,p \to \mu\,e$ in the case of flavour universal couplings to electrons and muons for the $R_2$ Model.