# Peer review of "Status of leptoquark models after LHC Run-2 and discovery prospects at future colliders"

_SciPost Physics_

## Round 1 · Referee Report · Anonymous (Referee 1) · 2024-7-3

Report

The paper provides a status of leptoquark models which is not up to date in two respects: first the motivations from B physics should be updated as they seem more than a year old. Indeed for example the Bs -> mu mu is indicated as not consistent with the SM, but taking both CMS and LHCb data into account it is in overall agreement with the SM at present. I suggest the authors update this part using recent fits. Second the choice of models for the leptoquarks is also obsolete and largely incomplete in many respects as still referring to papers mainly from 2016-2018. I suggest the authors to have a look for example to the recent papers of the Isidori group on leptoquark models consistent with flavour constraints and which also include an interesting UV completion.

Another main point concerns the bounds themselves: the analysis is a simple leading order comparison with data. More refined analyses exist in the literature. As an example see the NLO one for scalar leptoquarks at LHC in 2108.11404

In my opinion the paper does not provide an accurate present status of leptoquark models and misses existing work in the field which is more advanced and more up to date.

Recommendation

Ask for major revision

---

## Round 1 · Referee Report · Anonymous (Referee 2) · 2024-7-23

Strengths

The complete multiplet structure of the leptoquarks is taken into account whenver applicable.

Weaknesses

The results are results are not as general as indicated by the abstract and the conclusion but depend on the chosen coupling structures. No NLO corrections for the cross sections are taken into account.

Report

This paper does not meet at the current stage the expection requirements. Further details are required.

Attachment

Recommendation

Ask for major revision

---

## Round 1 · Referee Report · Anonymous (Referee 3) · 2024-7-24

Report

The paper aims to study bounds on LQ models on the basis of the still surviving anomalies in the B-sector. In doing so, the presented studies rely on the limits on the relevant Wilson coefficient of a low-energy weak Hamiltonian, derived in the literature at the end of 2022. However, it needs to be reiterated that it is questionable to what extent these exact limits are still valid. The CMS results (2401.07090) have been available as a PAS note for quite some time and it is not acceptable to entirely ignore them. In this context, the title “Status of leptoquark models after LHC Run2”, implying that all relevant Run2 results are taken into consideration, is simply misleading.

Furthermore, there is no discussion provided of what is already known in the literature regarding the impact of the 2022 LHCb measurement on the validity of leptoquark models and the values of the relevant Wilson coefficients. Such analysis have been done by multiple groups, not only Ref.7, but also e.g. 2212.10497, 2304.07330, 2310.05585. Given the amount of information available, the authors should justify why they pick a result from just one paper and how the results of other analyses might impact they conclusions.

Also, as the idea behind the presented studies is not very original (similar ideas, just differently realised, have been explored), I would expect that the authors would make an effort to make detailed studies, taking into account how the end results can be affected. In contrast, the presented analysis is very arbitrary: assumptions are made without discussing their validity or their effects on the conclusions, higher-order effects are only sometimes included, sometimes not, the limitations of or uncertainties on the derived bounds are not discussed, etc.

Altogether, the paper gives an impression of an outdated draft (also manifesting through the lack of any references to 2023 or 2024 papers), instead of a high-quality final product.

Recommendation

Ask for major revision

---

## Editorial Decision

awaiting_resubmission